# Statistical Analysis of Table-Tennis Ball Trajectories

**Ralf Schneider [1],\*, Lars Lewerentz [1], Karl Lüskow [1], Marc Marschall [1] and Stefan Kemnitz [2]**

[1]  Institute for Physics, University of Greifswald, 17498 Greifswald, Germany; lleweren@ipp.mpg.de (L.L.); lueskow@physik.uni-greifswald.de (K.L.); marc@marschalls.net (M.M.)

[2]  Institute of Computer Science, University of Rostock, 18051 Rostock, Germany; sk1013@uni-rostock.de

\*  Correspondence: schneider@uni-greifswald.de; Tel.: +49-3834-420-1400

**Abstract:** In this work, the equations of motion for table-tennis balls were numerically solved on graphics processing units (GPUs) using Compute Unified Device Architecture (CUDA) for systematical statistical studies of the impact of ball size and weight, as well as of net height, on the distribution functions of successful strokes. Half a billion different initial conditions involving hitting location, initial spin, and velocities were analyzed to reach sufficient statistical significance for the different cases. In this paper, an advanced statistical analysis of the database generated by the simulation is presented.

**Keywords:** rule changes; GPU computing; statistical analysis

## 1. Introduction

The medial appeal of table tennis seems to go down in terms of television (TV) hours, at least outside Asia. One of the reasons is the fact that the speed of the game is currently so high that it is very hard for spectators to follow the balls [1,2]. Possible counteractions to slow the game down involve using bigger balls or higher nets.

One intention of possible rule changes is to reduce the impact of spin on the game. Another goal is to reduce the speed of the balls to allow better visual tracking during the rallies [2]. Some rule changes, such as the use of a larger ball, different counting system, stricter limits for rubbers, or new service rules, were already implemented and new modifications are under discussion [2]. For the players, all rule or technical changes have strong impacts on their techniques and strategies, usually requiring adaptations of their individual training programs. Therefore, players are rather hesitant to new rules.

The 40-mm ball played today is 2 mm larger and 0.2 g heavier than the 38-mm ball used before. It has a larger air drag due to its larger cross-sectional area reducing the maximum velocities [3]. The mass distribution of the larger ball is shifted further away from the center compared with the 38 mm ball. This creates a larger inertial moment and reduces the spin. The larger 40-mm ball results in a velocity and spin reduction of about 5% to 10% [4,5]. However, the larger ball had practically no impact on the characteristics of table tennis, because larger exertions of forces by the players compensated for the effects of the size increase [4,6]. Because of the modified technique, the fitness of the individual player got more important. In modern table tennis, the forces for a stroke are created not only by the arms but also support from the whole body. A stronger athletic body allows more pronounced use of the legs producing larger forces on the ball, which are needed to compensate for the size increase. In addition, the wrist has to be used more effectively to produce spin. For the larger ball, only the use of the forearm is no longer sufficient for spin, as was the case for the 38-mm ball. The needs for larger exertion of forces amplify possible technical mistakes, because the individual movement execution gets extended [7].

One obvious strategy for reducing the maximum velocity in table-tennis rallies is to increase the net height. However, such a change will have a severe impact on the characteristics of table tennis,

because this will very directly limit fast spins, shots, and service. Therefore, until now, this change of rule was avoided and ball size was the preferred correction action. Nevertheless, a scientific database is still missing for a decision.

Usually, an empirical approach is followed to study the effect of such changes on the players and the game. Rather short and limited tests with some players are evaluated [8]. These tests are biased, because the players use the technique optimized for the existing situation (ball size, ball weight, and net height) and modifications needed for the new situation take too long for the players to be automatized in their training to be considered. In this work, the impact of larger balls or higher nets on table tennis trajectories was studied using computer simulations. A database was created to quantify the influence of such changes. Modifications in technique, tactics, strength, and fitness were not considered in this analysis. For a huge number of initial conditions, the effect on successful strokes was studied. This delivers the maximum number of possible strokes for different conditions in terms of statistical distributions. This already represents the best possible adaptation to the changes, independent of what this would mean for the players in terms of changes in their training. Half a billion different initial conditions involving hitting location, initial spin, and velocities were analyzed to reach sufficient statistical significance.

In our previous work [9], the distribution functions of successful hits were empirically compared for different cases with respect to their variations for the different variables, summing up the distributions with respect to all other variables. In this work, a more rigorous statistical analysis of the database is presented to identify the most significant variables. The multi-dimensional distribution functions of these variables were then used to identify the major differences for the different cases, thereby avoiding a possible bias in the analysis by correlation. An obvious example for a direct correlation between variables is the values of starting velocity and kinetic initial energy.

After a short discussion of the effects of larger balls and higher nets as measures to slow down table tennis, the forces acting on a moving ball are introduced. The very fast graphics processing unit (GPU) Compute Unified Device Architecture (CUDA) [10] computer code solving the equation of motion is described, and statistical analysis of trajectory distribution functions for different balls and net heights is presented. Finally, the results are summarized and discussed.

## 2. Materials and Methods

For a quantitative analysis of the effects of ball size and net height, a computational approach was followed. The basic element of the simulation was the solution of the equation of motion for table-tennis balls. The equation of motion needs a mathematical description of the acting forces. The gravitational force of the earth and aerodynamic forces determine the flight trajectory of a table-tennis ball. The gravitational force,

$$\vec{F_G} = m\vec{g} \tag{1}$$

alone results in a parabolic trajectory. This force acts toward the center of the earth and depends on the mass $m$ of the ball and the gravitational constant $g$ (9.81 m/s$^2$). The aerodynamic forces modify the simple parabola by air drag and lift. Air drag acts as friction force against the direction of the movement of the ball. A simple example for this force is the back-pushing of a hand being held out of a driving car. A larger velocity gives stronger force acting against the direction of the car. This force also gets larger if one puts out not only a part of the hand, but the full hand. It scales with the cross-sectional area. The mathematical expression is as follows:

$$\vec{F_D} = -\frac{1}{2}C_D\rho A v \vec{v} \tag{2}$$

where $\rho$ is the density of air, $A$ is the cross-sectional area for a ball with radius $r$ ($A = r^2\pi$), $v$ is the ball velocity, and $C_D$ is an air drag coefficient, which can be measured, e.g., in wind-tunnel experiments.

The second important aerodynamic force is the airlift. The so-called "Magnus effect", named after its discoverer Heinrich Gustav Magnus (1802–1870), is the reason that a rotating ball experiences a deviation from its flight path. The Magnus effect is a surface effect, because, around the spinning ball, a co-rotating air layer is formed at the surface of the ball. The flying and spinning ball induces a pressure imbalance, because, on one side, the ball is rotating with the airflow created by the movement of the ball in the air, while the other side is opposite to it. On the side where counter-rotation exists, the total velocity of the airflow is reduced, because both velocities compensate for each other partly. On the co-rotation side, a larger flow velocity is created, because both velocities add up. Higher velocity in a flow means lower pressure, and the pressure differences of the two sides lead to the deviating Magnus force, mathematically expressed with an airlift coefficient $C_L$ as

$$\vec{F_L} = \frac{1}{2}C_L\rho A v \vec{e_\omega} \times \vec{v} \tag{3}$$

The airlift force acts perpendicular to the axis of rotation $\vec{e_\omega}$ and to the velocity $\vec{v}$. Air drag and lift coefficients of a rotating ball (Figure 1) as a function of the ratio of spinning velocity to translational velocity were implemented into the computer code as a fit of experimental data [11–15].

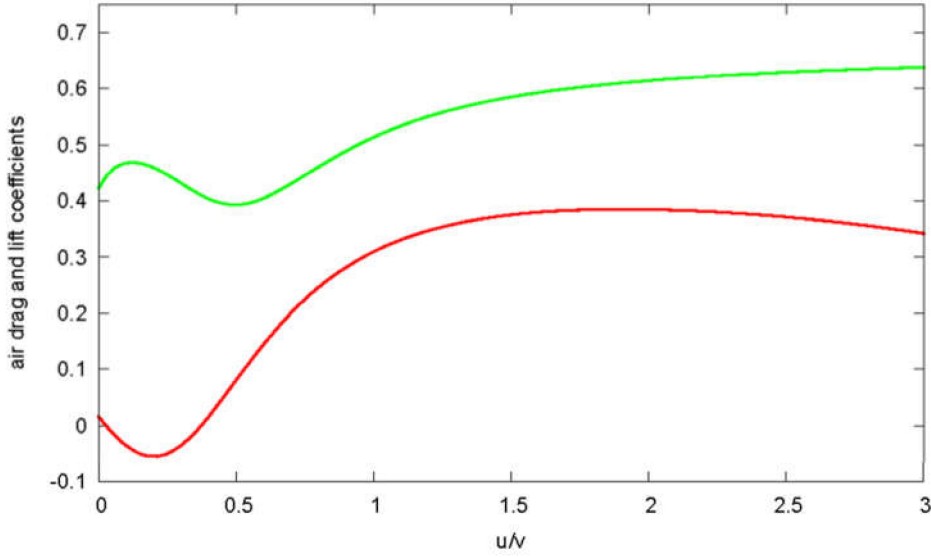

**Figure 1.** Air drag coefficient $C_D$ (upper green curve) and airlift coefficient $C_L$ (lower red curve) as a function of the ratio of spinning velocity $u$ to the translational velocity $v$.

During a topspin shot with forward rotation, the lift force acts downward; during a backspin with backward rotation, it acts upward.

Swirling balls, often quoted in soccer and volleyball, can be created when the ball is hit with a critical velocity leading to access of the inverse Magnus effect. It shows up in Figure 1 for low spinning velocities as a negative value of the airlift coefficient. This can lead also to swirling balls in table tennis, because, during the flight path, the regime of positive and negative airlift coefficients can change, resulting in swirling. However, for table-tennis balls, negative airlift coefficients exist only where the coefficient itself is already quite small. Therefore, the effect exists, but gives only deviations of some millimeters. The frequently quoted swirling balls with long pimples are, therefore, more a psychological effect than one due to physics: the pre-programmed movement of the player anticipates a flight path of a strongly rotating ball from a normal rubber sponge. The balls from long pimples with reduced rotation have a different flight path with less lift and fall down earlier such that the player misses the ball, and he complains that the ball was swirling.

The computer code solves the equation of motion of table-tennis balls for given initial positions, velocities, and spins (see one example in Figure 2). A Euler solver was used, because its algorithmic simplicity allowed an easy transfer onto the GPU with CUDA.

The implicit assumption of the Verlet algorithm (with or without velocity calculation) is that the driving force is conservative, i.e., on a closed path, no work is done. For our system studied here, this was not the case, since the drag and the Magnus force both influence the trajectory depending on the actual velocity. If a particle on a closed path comes back to its origin, the drag will slow it down. Energy is not conserved. In the standard Verlet scheme, the second force evaluation depends only on the position, whereas the force on the table-tennis ball is velocity-dependent. Thus, valid options are the Euler and Runge–Kutta (RK) methods. The RK fourth-order method requires four force calculations compared to one for the Euler method. Each force evaluation calls a function to calculate the drag coefficient and another one for the lift coefficient, because these depend on the actual translational velocity. These are rational functions where the numerator and denominator are polynomials of the third order. This makes every force evaluation costly.

For a very long flight of 2 s, a test trajectory with the standard time step of the code ($10^{-4}$ s) led to an end position even beyond the table. The difference of the final positions was 0.115mm for the Euler and RK4 methods. This was well below the precision of 1 mm we were looking for. However, the RK4 method was 4.37 times slower. Increasing the time step by a factor of five to compensate for the reduced calculation speed compared with the Euler method increased the error for the RK4 method beyond 1 mm, which we defined as our resolution limit.

A more extensive comparison of the Euler and RK4 algorithms was done for $10^5$ random test trajectories with identical start conditions for both integrators. The differences are listed in Table 1.

**Table 1.** Absolute values of the differences between Euler and RK4 integrators for the $10^5$ random test trajectories.

|  | Averages | Maximum | Mean-Square Deviation |
|---|---|---|---|
| $x$ (m) | $6.360759 \times 10^{-5}$ | $2.800000 \times 10^{-4}$ | $5.157352 \times 10^{-5}$ |
| $y$ (m) | $4.904501 \times 10^{-4}$ | $1.120000 \times 10^{-3}$ | $1.785499 \times 10^{-4}$ |
| $z$ (m) | $4.464907 \times 10^{-12}$ | $2.557150 \times 10^{-11}$ | $3.791575 \times 10^{-12}$ |
| $v$ (m/s) | $3.767089 \times 10^{-4}$ | $2.600000 \times 10^{-3}$ | $3.074972 \times 10^{-4}$ |
| $v_x$ (m/s) | $7.743345 \times 10^{-5}$ | $1.300000 \times 10^{-3}$ | $8.600171 \times 10^{-5}$ |
| $v_y$ (m/s) | $4.592264 \times 10^{-4}$ | $2.800000 \times 10^{-3}$ | $3.087756 \times 10^{-4}$ |
| $v_z$ (m/s) | $7.391139 \times 10^{-4}$ | $1.330000 \times 10^{-3}$ | $2.169369 \times 10^{-4}$ |

The results confirm the single test trajectory results. The number of successful trajectories and their distribution is nearly indistinguishable for the two cases and the 4.4-times-faster Euler integrator can be used.

One important factor for the limits of the time step for the integrator is the additional dependence of the drag and lift coefficients on the velocity (see Figure 1). These are non-linear terms in velocity, and changes in velocity in one time step need to be small enough to linearize this term during the time integration. Adaptive time-stepping could solve the problem, but this counteracts the optimized usage of GPUs, because this is only effective if the calculation effort for the different trajectories calculated in parallel is balanced. A strong variation of calculation time would reduce the effectivity of this parallelization strongly, because the slowest trajectory is then defining the limit. Therefore, the Euler method was chosen as a fast method for the calculation of half a billion trajectories for each case within the precision limit of 1 mm for the time step of $10^{-4}$ s.

For a statistical analysis of the effects of ball sizes and net heights on trajectories of table-tennis balls a Monte Carlo procedure was used.

Many different initial conditions were solved: $x$ was varied between 0.3 m and $-3$ m, representing hitting locations from 30 cm above the table to 3 m behind the table; $y$ was kept constant at 0.381 m, which is $\frac{1}{4}$ of the width of the table-tennis table. This was chosen as a representative position, the exact

location of the hitting point in $y$ (forehand or backhand position) is not important for this numerical test. Initial height $z$ was sampled from 0.4 m to $-0.4$ m. The direction of the initial velocity was determined in the following way: the horizontal angle was sampled between the limiting angles of the starting point to the net posts, and the elevation angle was chosen randomly. The spin axis (expressed as normalized spin vector) was also sampled randomly, which means topspin, backspin, and sidespin were included. The spinning of the ball is constant during the flight, as proven experimentally [16].

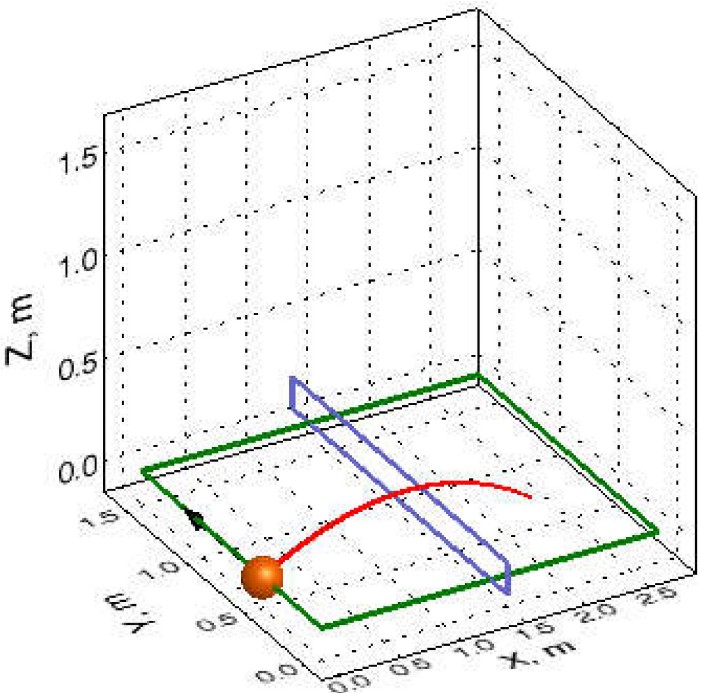

**Figure 2.** Three-dimensional (3D) trajectory of a table-tennis ball.

Monte Carlo studies using random numbers were done for the 38-mm ball with a weight of 2.5 g, used in tournaments until end of 2000, the actual 40-mm ball with 2.7 g, and a 44 mm-ball with a weight of 2.3 g, which was previously tested in Japan. For the 40-mm ball, an increase in net height of 1 and 3 cm was also analyzed.

The analysis was aimed particularly at fast shots. Therefore, only balls passing the net within 30-cm height distance were accepted. The absolute values of the translational velocities were limited from 20 to 200 km/h, the spinning velocities from 0 to 150 turns/s (which is equal to 9000 turns/min). These values were previously determined empirically as limits for 38-mm balls [17]. A ball is counted as a successful ball if it passes the net within the height limit and hits the other side of the table-tennis table.

The sampling of such a large number of initial conditions guarantees covering all possible combinations of initial parameters (positions, and translational and spinning velocities) for the different cases creating a successful stroke. Clearly, for different balls and net heights, the parameter space of initial conditions leading to successful strokes will be different. The database created in this study allowed also an analysis of this effect.

For each case, $5 \times 10^8$ initial conditions were sampled and trajectories calculated. Initially this was done on a Linux cluster with 32 cores. The run-time for each core was 20 h, resulting in a total run time of 640 h. Alternatively, GPU computing with CUDA was used on a Dell Precision T7500 desktop with NVIDIA Quadro FX3800. Here, only 3 h for the same calculation was needed. CUDA [8] is a programming interface using the parallel architecture of NVIDIA GPUs for general-purpose computing. CUDA library functions are provided as extensions of the C language, which allows for convenient and rather natural mapping of algorithms from C to CUDA. A compiler generates executable code for the

CUDA device. The central processing unit (CPU) identifies a CUDA device as a multi-core coprocessor. For the programmer, CUDA consists of a collection of threads running in parallel. A collection of threads, which is called a block, runs on a multiprocessor at a given time. The blocks form a so-called grid. They divide the common resources, such as registers and shared memory, equally among them. All threads of the grid execute a single program called the kernel. All memory available on the device can be accessed using CUDA with no restrictions on its representation. However, the access times vary for different types of memory. Shared and register memory are the fastest, as they locate on the multiprocessor (on chip). The shared memory has the lifetime of the block and it is accessible by any thread on the block from which it was created. This enhancement in the memory model allows programmers to better exploit the parallel power of the GPU for general-purpose computing. Additionally, the texture memory that is off-chip allows for faster reading compared to global memory due to caching.

Our implementation consists of two main procedures [8]. Firstly, a predefined number of trajectories were initialized on the CPU side. Thereafter, the ball movements were implemented on the GPU. One step of the equation of motion for the ball's trajectory, which includes the speed and the position of the ball, was computed in a kernel. The input parameter of the kernel function was the previous trajectory point. The calculations ran for a maximal number of iterations. In each iteration step, the updates of the ball's position and velocity were computed if the trajectory was not stopped earlier, e.g., when the ball flew beyond the table. The algorithm was optimized for GPU coding by minimizing the control flow, which slows down the parallel process. All trajectories were followed for the same number of iterations and, if the ball flew beyond the table, a flag identifying this was set. This procedure guaranteed optimum performance because all trajectories required the same calculation time and the threads were nearly perfectly balanced. At the end of the GPU calculation, the results of the trajectories were pushed back to the CPU; further diagnostics, such as identifying successful hits, was done there.

For each case, half a billion initial conditions were sampled and trajectories were calculated on a GPU with CUDA [8]. This also defines the uniqueness of this work, because a sufficient resolution of the phase space was possible only with this optimized coding and the usage of a GPU. Calculation of trajectories in table tennis mostly concentrates on individual cases without statistical analysis [18]. New interest for the fast calculation of table-tennis trajectories is motivated by the research on robots [19,20] and for the programming of computer games [21]. The large modeling database established by this procedure was analyzed in this paper using statistical methods to identify, in more detail, the major differences for the different cases and their major dependencies.

## 3. Results

Datasets of successful hits for the different cases (38-mm ball, 40-mm ball, 44-mm ball, and 40-mm ball with 1- and 3-cm-higher net) were created. Table 2 shows that, in terms of successful strokes, the 38-mm and 40-mm balls varied only marginally. The heavier 44-mm ball allowed for more strokes that were successful, whereas the higher net reduced this number.

**Table 2.** Number of successful trajectories for the different cases.

| Case | Number of Successful Trajectories |
| --- | --- |
| 38 mm | 2,795,262 |
| 40 mm | 2,793,202 |
| 44 mm | 3,282,767 |
| 40 mm + 1-cm net | 2,672,572 |
| 40 mm + 3-cm net | 2,470,891 |

The list of dataset variables is presented in Table 3.

**Table 3.** List of variables used in the database.

| Index | Variable | Comment |
|:---:|:---:|:---|
| 1 | $x_0$ | $x$-position of starting point (varied between $-3$ and 0.3 m) |
| 2 | $y_0$ | $y$-position of starting point (not varied; 0.381 m) |
| 3 | $z_0$ | $z$-position of starting point (varied between $-0.4$m and 0.4 m) |
| 4 | $v_{start}$ | absolute value of start velocity (between 20 km/h and 200 km/h) |
| 5 | $v_{x;0}$ | $x$-component of start velocity according to start angles |
| 6 | $v_{y,0}$ | $y$-component of start velocity according to start angles |
| 7 | $v_{z,0}$ | $z$-component of start velocity according to start angles |
| 8 | $x_e$ | $x$-position of ball hitting the table |
| 9 | $y_e$ | $y$-position of ball hitting the table |
| 10 | $z_e$ | $z$-position of ball hitting the table (0 m) |
| 11 | $v_e$ | absolute value of the final ball velocity |
| 12 | $v_{x,e}$ | $x$-component of the final ball velocity |
| 13 | $v_{y,e}$ | $y$-component of the final ball velocity |
| 14 | $v_{z,e}$ | $z$-component of the final ball velocity |
| 15 | $\omega_0$ | absolute value of the spinning velocity (between 0 and 150 turns/s) |
| 16 | $\omega_x$ | $x$-component of the normalized spin vector |
| 17 | $\omega_y$ | $y$-component of the normalized spin vector |
| 18 | $\omega_z$ | $z$-component of the normalized spin vector |
| 19 | $E_{kin,0}$ | initial kinetic energy |
| 20 | $E_{kin,e}$ | final kinetic energy |
| 21 | $E_{rot}$ | rotational energy |
| 22 | $z_{over\ net}$ | height of the ball over the net for a successful stroke |
| 23 | $z_{max}$ | maximal height during a successful stroke |

The data of the different cases were loaded into a Pandas [22] table. The data of the trajectories themselves were normalized, i.e., the mean was zero and the standard deviation was set to one. The two variables $y_0$ and $z_e$ had to be removed from the original data, since their standard deviation was practically zero. The initial $y$-position $y_0$ was not varied and $z_e$ was the height at the end of the trajectory, which was, by definition, zero for successful trajectories. For these datasets, the principal component analysis (PCA) from the Python package scikit-learn [23] was applied to the 21-dimensional parameter space. Table 4 shows the results for the 40-mm ball reference case; the other PCA results can be found in Appendix A.

**Table 4.** Principal component analysis (PCA) for the full set of variables: 40-mm ball reference case.

| | Standard Deviation | Proportion of Variance | Cumulative Proportion | Eigenvalue |
|:---|:---:|:---:|:---:|:---:|
| PC1 | 2.700487 | 0.347268 | 0.347268 | 7.292632 |
| PC2 | 1.633091 | 0.126999 | 0.474267 | 2.666987 |
| PC3 | 1.508682 | 0.108387 | 0.582654 | 2.276124 |
| PC4 | 1.401304 | 0.093507 | 0.676161 | 1.963654 |
| PC5 | 1.111800 | 0.058862 | 0.735023 | 1.236100 |
| PC6 | 1.011844 | 0.048754 | 0.783777 | 1.023829 |
| PC7 | 1.001355 | 0.047748 | 0.831525 | 1.002711 |
| PC8 | 0.998771 | 0.047502 | 0.879027 | 0.997544 |
| PC9 | 0.994223 | 0.047070 | 0.926098 | 0.988479 |
| PC10 | 0.906493 | 0.039130 | 0.965228 | 0.821729 |
| PC11 | 0.521662 | 0.012959 | 0.978186 | 0.272132 |
| PC12 | 0.469405 | 0.010492 | 0.988679 | 0.220341 |
| PC13 | 0.309721 | 0.004568 | 0.993247 | 0.095927 |
| PC14 | 0.235542 | 0.002642 | 0.995889 | 0.055480 |
| PC15 | 0.190701 | 0.001732 | 0.997620 | 0.036367 |
| PC16 | 0.155887 | 0.001157 | 0.998778 | 0.024301 |
| PC17 | 0.128646 | 0.000788 | 0.999566 | 0.016550 |
| PC18 | 0.082267 | 0.000322 | 0.999888 | 0.006768 |

**Table 4.** *Cont.*

|       | Standard Deviation | Proportion of Variance | Cumulative Proportion | Eigenvalue |
|-------|-------------------|------------------------|-----------------------|------------|
| PC19  | 0.035563          | 0.000060               | 0.999948              | 0.001265   |
| PC20  | 0.029352          | 0.000041               | 0.999989              | 0.000862   |
| PC21  | 0.015071          | 0.000011               | 1.000000              | 0.000227   |

Many components were superfluous, and their contribution was negligible. The first strong drop of eigenvalues can be seen in Figure 3 after four components, and a second step appeared after 10 components. The first four components accounted for about 68% of the variance. This subspace could be used to identify the major differences. An optimum set for the four remaining variables was found with vstart: $\omega_0$, $\omega_y$, and $\omega_z$. This combination led to a nearly equal distribution of eigenvalues, as shown in Table 5.

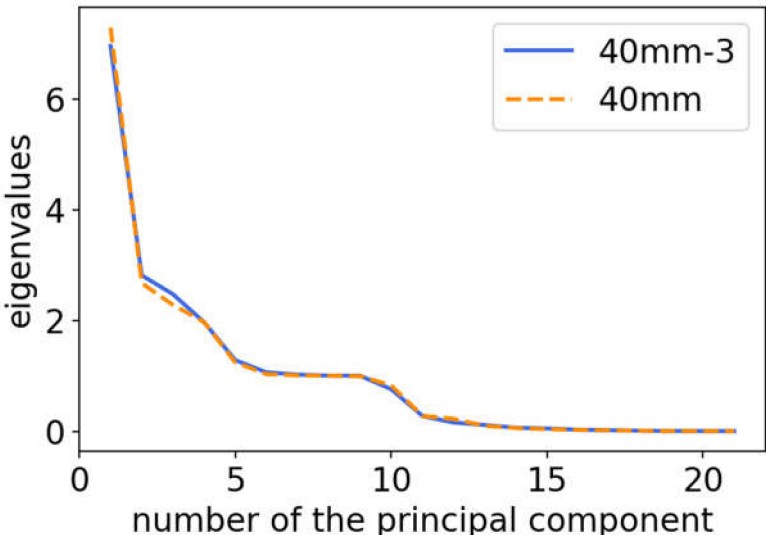

**Figure 3.** Eigenvalues as a function of each principal component for the databases of the 40-mm ball and the 40-mm ball with 3-cm-higher net.

A further reduction would result in a severe loss of variance in the data. Using more variables would lead to a strong drop of the next eigenvalue, demonstrating the reasonable limit of variables to this subset.

**Table 5.** PCA with a reduced set of variables for the 40-mm case.

|     | Standard Deviation | Proportion of Variance | Cumulative Proportion | Eigenvalue |
|-----|-------------------|------------------------|-----------------------|------------|
| PC1 | 1.007513          | 0.253771               | 0.253771              | 1.015083   |
| PC2 | 1.001774          | 0.250888               | 0.504658              | 1.003551   |
| PC3 | 0.997023          | 0.248514               | 0.753172              | 0.994055   |
| PC4 | 0.993636          | 0.246828               | 1.000000              | 0.987312   |

This reduction in the number of variables to four was plausible because the start velocity, together with the rotation velocity and the rotation axis, determines most of the dataset variation. Only two components of the rotation axis were needed, because this was a unit vector and, from two components, the third component was pre-determined. The influence of starting positions and initial angles of the start velocity was much weaker and was reflected by the second drop in eigenvalues after the 10th component.

Using this subset of variables, a more detailed statistical analysis was then done. Comparing the results for eigenvalues and eigenvectors of the 38-mm ball and 40-mm ball databases gave nearly

identical results (Figure 4). The first eigenvector was dominated by the contribution from the spin velocity, the second one by the translational velocity, the third one by the topspin component, and the fourth by the sidespin.

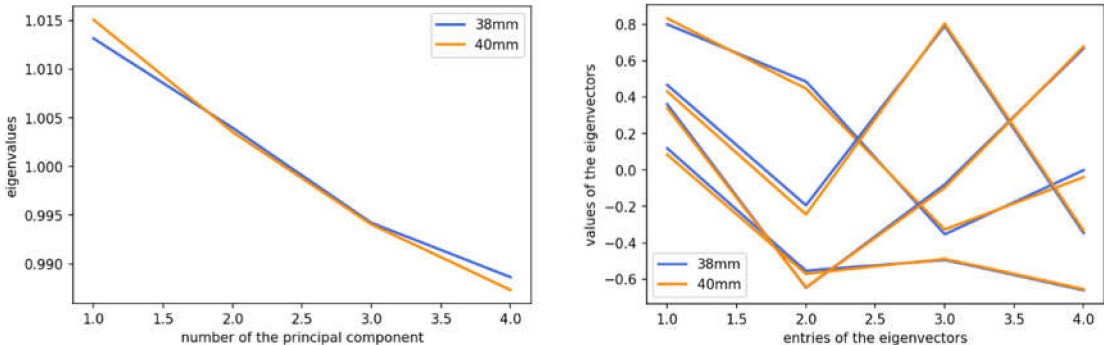

**Figure 4.** (**left**) Eigenvalues as a function of each principal component for the databases of the 38-mm ball and the 40-mm ball. (**right**) Values of the eigenvectors for the two cases.

To analyze the differences between the different cases a four-dimensional histogram of each case was calculated with the Numpy "histogramdd" routine [24]. The relative deviation *d* was calculated with respect to the case of the 40-mm ball using the following equation:

$$d_{i,j}^{case,40\text{mm}} = \sum_{k,l} \frac{\Phi^{case} - \Phi^{40\text{mm}}}{\Phi^{40\text{mm}} + 1}, \tag{4}$$

where $\Phi$ is the number of trajectories within the four-dimensional bin with the coordinates $(i, j, k, l)$ of the histogram; $\Phi$ is an abbreviation for $\Phi_{i,j,k,l}$. Two dimensions from $v_{start}$, $\omega_0$, $\omega_y$, and $\omega_z$ were selected for the plots denoted by *i* and *j*, whereas the relative deviation was summed up over the remaining dimensions, *k* and *l*. In order to account only statistically sensible data, differences in the numerator less than 10 were neglected. Plots were then generated with the Python packages Matplotlib [25] and HoloViews [26].

The 38-mm ball and the 40-mm ball were practically identical (Figure 5); very small differences showed up close to statistical variation limits.

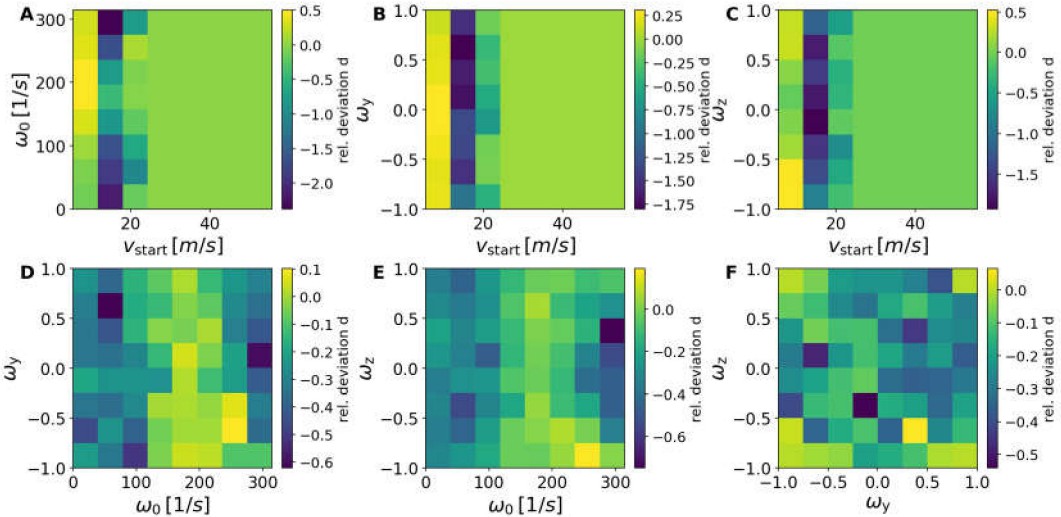

**Figure 5.** The two-dimensional (2D) distribution plots of the relative deviations as defined in the text of the databases for the 38-mm ball and for the 40-mm ball.

The larger and heavier 44-mm ball showed clear deviations from the 40-mm reference database (Figures 6 and 7).

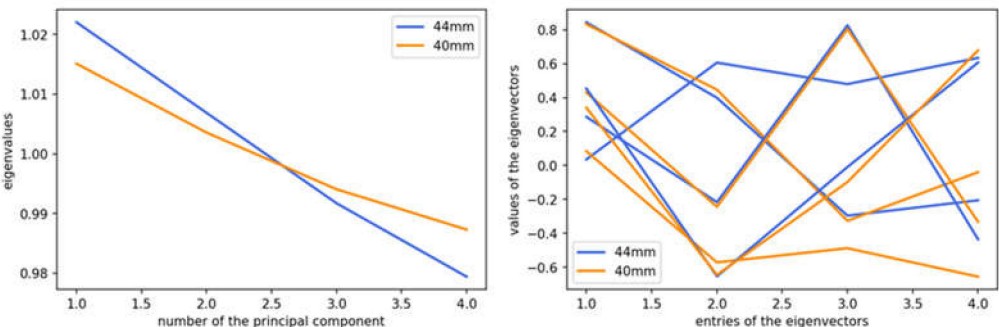

**Figure 6.** (**left**) Eigenvalues as a function of each principal component for the databases of the 44-mm ball and the 40-mm ball. (**right**) Values of the eigenvectors for the two cases.

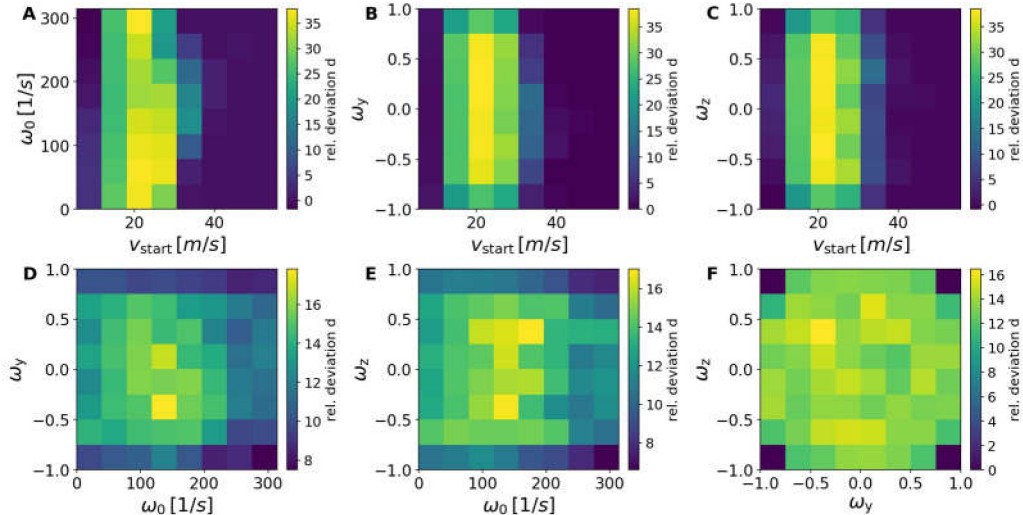

**Figure 7.** The 2D distribution plots of the relative deviations as defined in the text of the databases for the 44-mm ball and for the 40-mm ball.

The first eigenvector had an even stronger weighting of the spin compared to the 40-mm reference case. Also, the second and third eigenvectors increased the dominant weights for the translational velocity and the topspin, whereas the fourth eigenvector stayed unchanged.

This change in eigenvectors can be easily understood. The global increase of successful strokes for the 44-mm ball compared to the 40-mm ball resulted mainly from the fact that the 40-mm ball was heavier (increasing the gravitational force).

One needs larger spins to influence the trajectories of the 44-mm ball. For higher initial velocity, a larger number of successful shots was possible with the 44-mm ball, because the effect of drag forces increased with larger size.

Deviations for the cases with 1- and 3-cm-higher nets for the 40-mm balls compared with the normal 40-mm scenario were more pronounced.

With an increased net height of 1 cm (see Figures 8 and 9) and 3 cm (see Figures 10 and 11) for the 40-mm ball, a strong reduction in the number of successful strokes was detected. The reduction increased, as naively expected, when the net was higher.

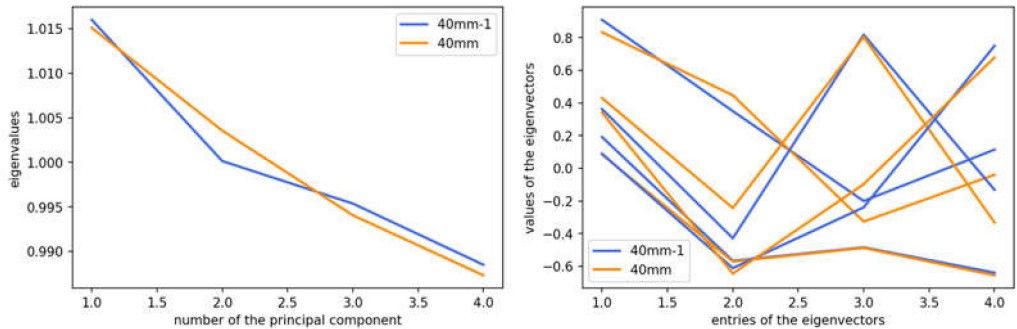

**Figure 8.** (**left**) Eigenvalues as a function of each principal component for the databases of the 40-mm ball with 1-cm-higher net and the 40-mm ball for standard net height. (**right**) Values of the eigenvectors for the two cases.

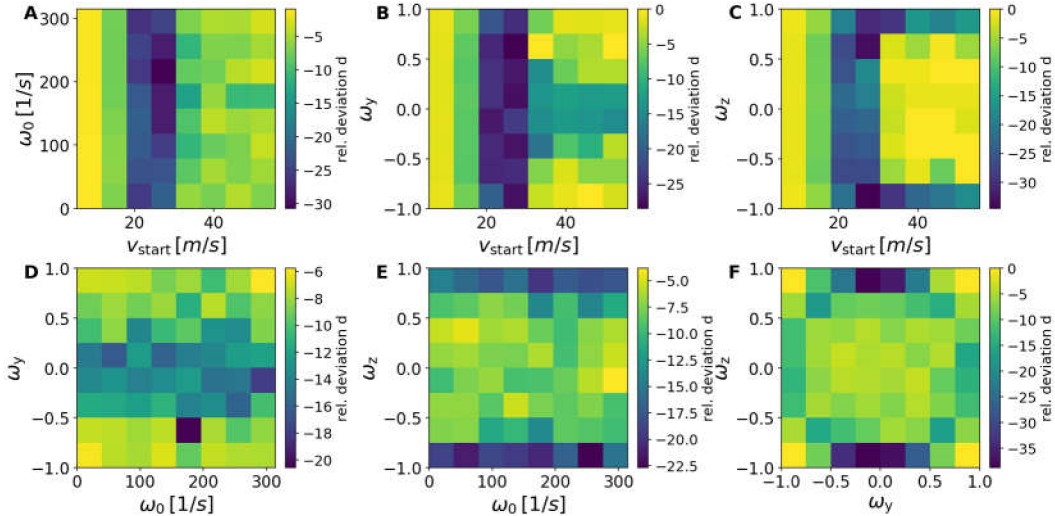

**Figure 9.** The 2D distribution plots of the relative deviations as defined in the text of the databases for the 40-mm ball with 1-cm-higher net and for the 40-mm ball with standard net height.

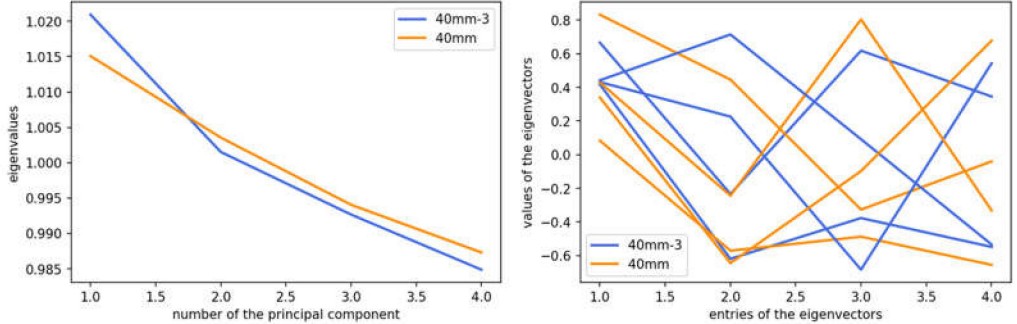

**Figure 10.** (**left**) Eigenvalues as a function of each principal component for the databases of the 40-mm ball with 3-cm-higher net and the 40 mm ball for standard net height. (**right**) Values of the eigenvectors for the two cases.

The first eigenvector had larger contributions from spin, but now also significant contributions from translational velocity. The second eigenvector had a reduced weighting of the translational velocity, but a larger weighting of the sidespin. The third eigenvector increased also the influence of sidespin, whereas the fourth eigenvector showed an increase of the spin contribution with a further reduction of the translational velocity. This was the result of the increased net height. If the starting velocity was too low, the ball was not able to pass the higher net successfully. Therefore, a reduction of

successful strokes appeared for low starting velocities compared to the standard net. The higher net needed stronger spins and reduced velocities for successful strokes. For very low velocities, the impact of the air drag was not yet important, resulting in a smaller reduction compared to higher velocities of about 30 m/s (or higher), where a strong reduction is observed. This reduction in the number of successful trajectories is equivalent to a slowing-down of the game.

The strong reduction in the number of successful strokes was linked with rather small top spin components $\omega_y$, but rather larger sidespin components $\omega_z$.

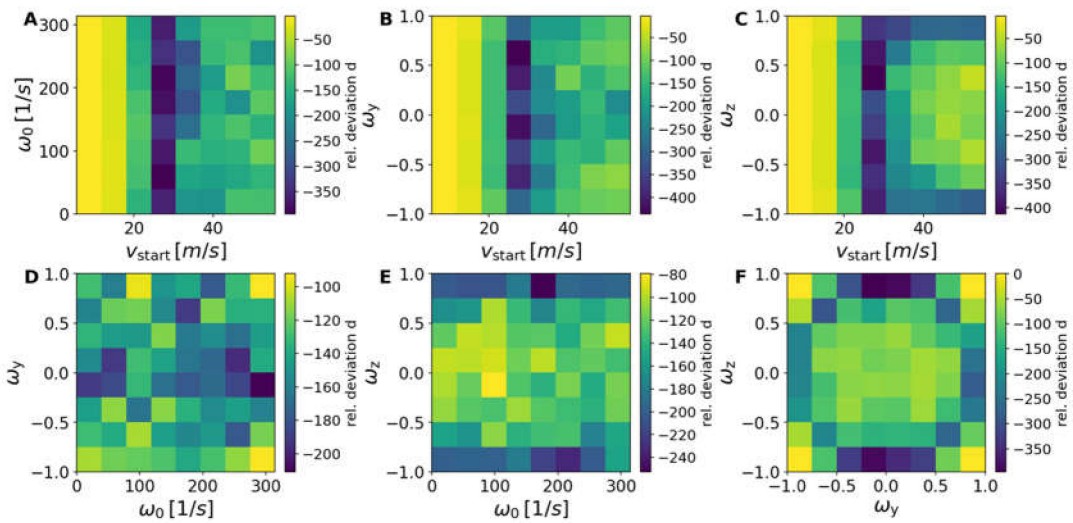

**Figure 11.** The 2D distribution plots of the relative deviations as defined in the text of the databases for the 40-mm ball with 3-cm-higher net and for the 40 mm ball with standard net height.

In addition, cluster analysis was applied to the data in order to dissect the properties of the most interesting cases. The datasets of the trajectories included roughly three million entries. A naive approach with an agglomerative clustering algorithm would easily exceed the memory of most computer systems. The agglomerative clustering with no constraints scaled quadratically in memory such that a crude estimate amounted to 36 TB = $(3 \times 10^6)^2 \times 4$ B.

Therefore, balanced iterative reducing and clustering using hierarchies (BIRCH) clustering from the Python scikit-learn library [23] was used to reduce the number of data points to about 1100 points. This tool is rather memory-effective and designed for big datasets. The method generates a tree structure which can also be passed to other clustering algorithms. In our case, this was the agglomerative clustering from the same library [23]. The metric used was Ward linkage with Euclidean affinity. This linkage minimized the variance of the clusters being merged.

From these reduced datasets, dendrograms were generated with the Python scipy library [27]. A dendrogram is a plot of a hierarchical binary tree. The height of each line represents the distance between different datapoints with the given metric.

From the dendrograms (see Figure 12), one can visually deduce the number of clusters for each case. Cutting the tree at a given height with maximum variation of the metrics delivered the minimum number of clusters required for this case.

For all cases, one can find four clusters as sufficient to represent the largest metric variations. The reduced datasets for all cases were further diminished with K-means clustering from scikit-learn [23] until four clusters remained. The centers of the clusters contained the condensed properties of the different systems and could be compared easily. The sizes of the symbols also represent the number of members of the clusters, showing that they were nearly identical.

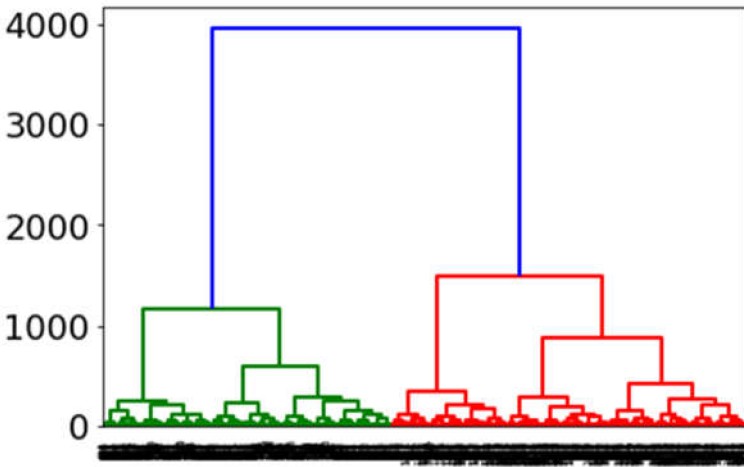

**Figure 12.** Dendrogram of the 40-mm reference case.

Figures 13–15 show the most significant differences for the cluster centers of the four clusters for the different cases. The larger 44-mm ball showed higher initial velocities for all four cluster centers compared to the 40-mm ball, as shown also in previous work [8], because the 40-mm ball was heavier, as discussed before (Figure 13).

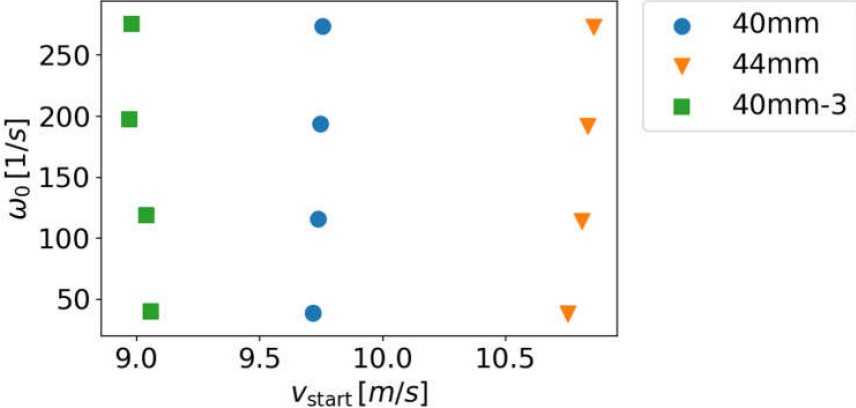

**Figure 13.** Cluster center coordinates of spin velocity versus start velocity for the four clusters for the different cases.

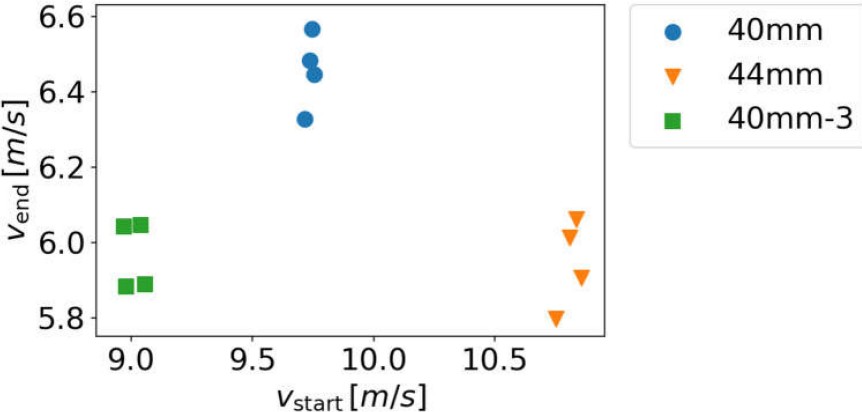

**Figure 14.** Cluster center coordinates of end velocity versus start velocity for the four clusters for the different cases.

Included in the cluster analysis, one can also see from Figure 14 shows that, even for the higher initial velocities in the 44-mm case, a clear slowing-down of the end velocity appeared. The same was obtained for the 40-mm case with increased net height (shown is the case with 3-cm increased height).

Larger $\omega_z$ and $\omega_y$ are visible in Figure 15 for the centers of the 40-mm case with increased net height and even stronger effects for the 44-mm ball, which had a smaller weight than the 40-mm ball. One needed larger spins to influence the trajectories of the 44-mm ball, because the effect of drag forces increased with larger size. The higher-net case needed stronger spins and reduced velocities for successful strokes. The $\omega_z$ values were larger than the $\omega_y$ contributions, meaning that stronger sidespin appeared, as discussed before.

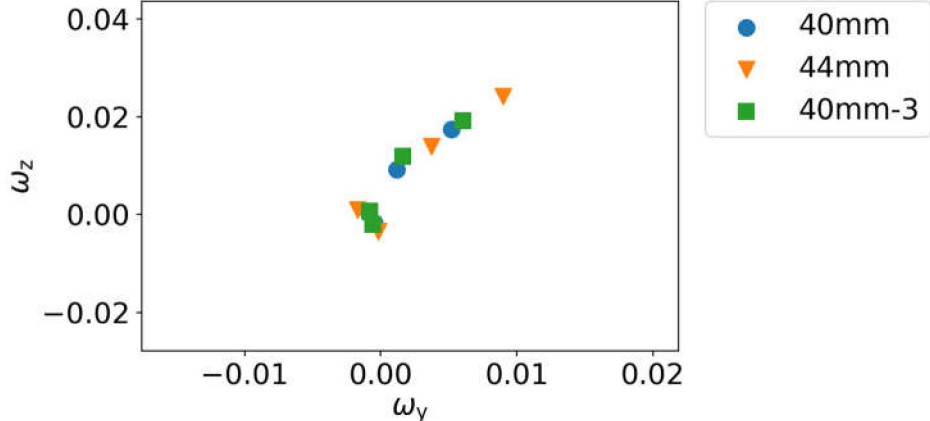

**Figure 15.** Cluster center coordinates of side spin component $\omega_z$ versus topspin component $\omega_y$ of the normalized spin vector for the four clusters for the different cases.

## 4. Discussion

The databases for the 38- and 40-mm balls showed only variations within statistical fluctuations for the multi-dimensional distribution of successful shots, whereas the 44-mm ball results were already significantly different compared to the 40-mm ball, as discussed also in Reference [8]. The analysis confirmed the empirical observation that the change of the ball in the year 2000 from a 38-mm to a 40-mm-ball could be compensated for such that their resulting trajectory distribution functions were nearly identical. This was achieved in reality by adaptations of technique and the material. One needed larger spins to influence the trajectories of the 44-mm ball with smaller weight compared with the 40-mm ball. For higher initial velocity, a larger number of successful shots was possible with the 44-mm ball, because the effect of drag forces increased with larger size. Very high velocities above 35 m/s were suppressed earlier for the 44-mm ball. A larger ball of 44 mm with small weight is one option for suppressing high velocities; however, the players could compensate for this by improving their physical fitness to perform stronger shots.

Most of the results of the previous empirical analysis are supported. A clear influence was visible for the 40-mm ball upon increasing the net height. For smaller initial and end velocities of about 10 m/s, a reduction in the number of successful trajectories showed up, being equivalent to a slowing-down of the game. For very low velocities, the impact of the air drag was not yet important, resulting in a larger number of successful trajectories. As a new result, the importance of stronger sidespin for the cases of higher nets was identified, which would change the characteristics of long rallies in table tennis. For these cases, not only are reduced velocities important, but the tactical possibilities will be modified by reducing the number of spinless short trajectories. The strong reduction in the number of successful strokes was linked with rather small top spin components $\omega_y$, but rather larger sidespin components $\omega_z$. This means that the game will not only slow down, but also diagonal play with longer reaction times for the opponent will get more important than fast parallel balls. One can also

expect from this longer and more attractive rallies. However, the characteristics of the game will change strongly, because the possibilities for successful trajectories are limiting technical and tactical alternatives, reducing especially the influence of service.

Modifications of basic rules of table tennis, such as ball size and net height, can reduce the maximum velocities; however, such modifications will be linked to severe changes in the characteristics of table tennis: dynamics, technique, and strategy will change strongly. The question is if a possible gain in attractiveness of table tennis for TV upon implementing such changes is worth the loss of key elements of existing table tennis.

## 5. Conclusions

Databases for trajectories of table-tennis balls of different sizes and different net heights were used for an analysis of the influence of such modifications on the game. The solution of the equation of motion for half a billion of different combinations of initial positions, velocities, and spins using a Euler solver on a GPU with CUDA allowed a good coverage of the operational space and the creation of such databases within reasonable run-time and computer resources. This is a new tool for the evaluation of effects of possible rule changes discussed for the slowing-down of table tennis to improve its medial appearance. It can replace the purely empirical approach followed so far, where rather short and limited tests with some players were done. These tests were always biased, because the players always used the technique optimized for the existing situation (ball size, ball weight, and net height) and modifications needed for the new situation took too long for the players to be automatized in their training to be considered.

Statistical analyses presented in this work (PCA and hierarchical cluster analysis) support the simplified analysis comparing distribution functions of successful shots done before. A slowing-down of the game would be possible upon using either a larger 44-mm ball or an increased net height (up to 3 cm) for the standard ball of 40 mm. A larger ball of 44 mm with small weight is one option for suppressing high velocities; however, the players could compensate for this by improving their physical fitness to perform stronger shots. As a new result, the importance of stronger sidespin for the cases of higher nets was identified, which would change the characteristics of long rallies in table tennis. For these cases, not only are reduced velocities important, but the tactical possibilities will be modified by reducing the number of spinless short trajectories. Diagonal play with longer reaction times for the opponent will get more important than fast parallel balls. One can also expect from this longer and more attractive rallies.

**Author Contributions:** Conceptualization, R.S.; methodology, R.S.; software, L.L., K.L., M.M., and S.K.; formal analysis, L.L.; writing—original draft preparation, R.S. and K.L.; writing—review and editing, R.S., L.L., K.L., and S.K.; visualization, L.L. and K.L.

**Funding:** This research received no external funding.

**Conflicts of Interest:** The authors declare no conflicts of interest.

## Appendix A

The PCA results for the different databases are presented here.

**Table A1.** PCA for the full set of variables: 38-mm ball case.

|  | Standard Deviation | Proportion of Variance | Cumulative Proportion | Eigenvalue |
|---|---|---|---|---|
| PC1 | 2.705554 | 0.348573 | 0.348573 | 7.320027 |
| PC2 | 1.633557 | 0.127072 | 0.475644 | 2.668508 |
| PC3 | 1.508753 | 0.108397 | 0.584041 | 2.276336 |
| PC4 | 1.400921 | 0.093456 | 0.677498 | 1.962581 |
| PC5 | 1.108199 | 0.058481 | 0.735979 | 1.228106 |
| PC6 | 1.008654 | 0.048447 | 0.784425 | 1.017382 |
| PC7 | 1.001265 | 0.047740 | 0.832165 | 1.002531 |
| PC8 | 0.998664 | 0.047492 | 0.879657 | 0.997331 |
| PC9 | 0.994456 | 0.047093 | 0.926749 | 0.988943 |
| PC10 | 0.906223 | 0.039107 | 0.965856 | 0.821241 |
| PC11 | 0.519211 | 0.012837 | 0.978693 | 0.269580 |
| PC12 | 0.474889 | 0.010739 | 0.989432 | 0.225520 |
| PC13 | 0.288775 | 0.003971 | 0.993403 | 0.083391 |
| PC14 | 0.235285 | 0.002636 | 0.996039 | 0.055359 |
| PC15 | 0.188227 | 0.001687 | 0.997727 | 0.035429 |
| PC16 | 0.154320 | 0.001134 | 0.998861 | 0.023815 |
| PC17 | 0.124096 | 0.000733 | 0.999594 | 0.015400 |
| PC18 | 0.079731 | 0.000303 | 0.999897 | 0.006357 |
| PC19 | 0.034417 | 0.000056 | 0.999953 | 0.001185 |
| PC20 | 0.028054 | 0.000037 | 0.999991 | 0.000787 |
| PC21 | 0.014093 | 0.000009 | 1.000000 | 0.000199 |

**Table A2.** PCA for the full set of variables: 44-mm ball case.

|  | Standard Deviation | Proportion of Variance | Cumulative Proportion | Eigenvalue |
|---|---|---|---|---|
| PC1 | 2.656949 | 0.336161 | 0.336161 | 7.059382 |
| PC2 | 1.625639 | 0.125843 | 0.462004 | 2.642702 |
| PC3 | 1.501421 | 0.107346 | 0.569350 | 2.254266 |
| PC4 | 1.402186 | 0.093625 | 0.662975 | 1.966126 |
| PC5 | 1.158213 | 0.063879 | 0.726854 | 1.341458 |
| PC6 | 1.028932 | 0.050414 | 0.777268 | 1.058701 |
| PC7 | 1.002442 | 0.047852 | 0.825120 | 1.004891 |
| PC8 | 0.998550 | 0.047481 | 0.872601 | 0.997102 |
| PC9 | 0.990976 | 0.046763 | 0.919365 | 0.982033 |
| PC10 | 0.900720 | 0.038633 | 0.957998 | 0.811297 |
| PC11 | 0.555513 | 0.014695 | 0.972693 | 0.308595 |
| PC12 | 0.439536 | 0.009200 | 0.981892 | 0.193192 |
| PC13 | 0.435854 | 0.009046 | 0.990939 | 0.189969 |
| PC14 | 0.250304 | 0.002983 | 0.993922 | 0.062652 |
| PC15 | 0.224921 | 0.002409 | 0.996331 | 0.050589 |
| PC16 | 0.177091 | 0.001493 | 0.997824 | 0.031361 |
| PC17 | 0.171912 | 0.001407 | 0.999232 | 0.029554 |
| PC18 | 0.105470 | 0.000530 | 0.999761 | 0.011124 |
| PC19 | 0.054237 | 0.000140 | 0.999901 | 0.002942 |
| PC20 | 0.040251 | 0.000077 | 0.999979 | 0.001620 |
| PC21 | 0.021195 | 0.000021 | 1.000000 | 0.000449 |

**Table A3.** PCA for the full set of variables: case for the 40-mm ball with a 1-cm-higher net.

|       | Standard Deviation | Proportion of Variance | Cumulative Proportion | Eigenvalue |
|-------|--------------------|------------------------|-----------------------|------------|
| PC1   | 2.673867           | 0.340455               | 0.340455              | 7.149567   |
| PC2   | 1.641592           | 0.128325               | 0.468780              | 2.694827   |
| PC3   | 1.534772           | 0.112168               | 0.580948              | 2.355527   |
| PC4   | 1.401452           | 0.093527               | 0.674475              | 1.964069   |
| PC5   | 1.122218           | 0.059970               | 0.734446              | 1.259375   |
| PC6   | 1.016426           | 0.049196               | 0.783642              | 1.033122   |
| PC7   | 1.002944           | 0.047900               | 0.831542              | 1.005897   |
| PC8   | 0.999019           | 0.047526               | 0.879067              | 0.998039   |
| PC9   | 0.996530           | 0.047289               | 0.926357              | 0.993072   |
| PC10  | 0.909107           | 0.039356               | 0.965713              | 0.826476   |
| PC11  | 0.524761           | 0.013113               | 0.978826              | 0.275374   |
| PC12  | 0.426978           | 0.008681               | 0.987507              | 0.182311   |
| PC13  | 0.339049           | 0.005474               | 0.992981              | 0.114954   |
| PC14  | 0.241137           | 0.002769               | 0.995750              | 0.058147   |
| PC15  | 0.195555           | 0.001821               | 0.997571              | 0.038242   |
| PC16  | 0.155493           | 0.001151               | 0.998722              | 0.024178   |
| PC17  | 0.130566           | 0.000812               | 0.999534              | 0.017048   |
| PC18  | 0.084288           | 0.000338               | 0.999872              | 0.007105   |
| PC19  | 0.037479           | 0.000067               | 0.999939              | 0.001405   |
| PC20  | 0.031239           | 0.000046               | 0.999986              | 0.000976   |
| PC21  | 0.017289           | 0.000014               | 1.000000              | 0.000299   |

**Table A4.** PCA for the full set of variables: case for the 40-mm ball with a 3-cm-higher net.

|       | Standard Deviation | Proportion of Variance | Cumulative Proportion | Eigenvalue |
|-------|--------------------|------------------------|-----------------------|------------|
| PC1   | 2.637615           | 0.331286               | 0.331286              | 6.957013   |
| PC2   | 1.676202           | 0.133793               | 0.465079              | 2.809653   |
| PC3   | 1.571444           | 0.117592               | 0.582671              | 2.469438   |
| PC4   | 1.400881           | 0.093451               | 0.676122              | 1.962469   |
| PC5   | 1.130068           | 0.060812               | 0.736934              | 1.277054   |
| PC6   | 1.028473           | 0.050369               | 0.787304              | 1.057757   |
| PC7   | 1.007619           | 0.048347               | 0.835651              | 1.015296   |
| PC8   | 0.999209           | 0.047544               | 0.883195              | 0.998420   |
| PC9   | 0.998222           | 0.047450               | 0.930645              | 0.996447   |
| PC10  | 0.867697           | 0.035852               | 0.966497              | 0.752898   |
| PC11  | 0.521240           | 0.012938               | 0.979435              | 0.271691   |
| PC12  | 0.394374           | 0.007406               | 0.986841              | 0.155531   |
| PC13  | 0.333126           | 0.005284               | 0.992125              | 0.110973   |
| PC14  | 0.246435           | 0.002892               | 0.995017              | 0.060730   |
| PC15  | 0.221226           | 0.002331               | 0.997348              | 0.048941   |
| PC16  | 0.163778           | 0.001277               | 0.998625              | 0.026823   |
| PC17  | 0.131183           | 0.000819               | 0.999445              | 0.017209   |
| PC18  | 0.087340           | 0.000363               | 0.999808              | 0.007628   |
| PC19  | 0.046263           | 0.000102               | 0.999910              | 0.002140   |
| PC20  | 0.037509           | 0.000067               | 0.999977              | 0.001407   |
| PC21  | 0.022068           | 0.000023               | 1.000000              | 0.000487   |

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
