# Peer review of "Statistical Analysis of Table-Tennis Ball Trajectories"

_applsci, doi:10.3390/app8122595_

Reviewer 1 Report

1.  The authors had the following claims.

A commonly used Runge-Kutta algorithm was not chosen, because it has larger computational costs. A fourth order Runge Kutta approach needs to calculate four times the forces, which slows down the code performance in our case compared to the simple Euler method.If the key objective of the paper is the accuracy of the proposed model, then the computational cost should not be a reason that prevents us from choosing a better algorithm. In other words, the authors can spend much more time in conducting experiments for higher accuracy if the Runge-Kutta algorithm is better than the Euler method.

2.  The authors claimed that “This was not compensated by the larger time step possible with the Runge-Kutta method compared to the Euler method.” Please conduct an experiment to prove the above claim and then accordingly present the results in the current version.

3. The authors had the following claims.The dependence of the aerodynamic forces on the velocity also does not allow the use of a Verlet algorithm. Therefore, we decided to stay with the Euler method.It is not clear why the dependence does not allow the use of a Verlet algorithm. Usually, dependence only influences the steps of how to implement the dependent algorithms. Please give a more detailed explanation.

4.The authors claimed that “The spinning of the ball is taken constant during the flight.” Is the assumption reasonable? Why? How does the assumption influence the results?

Author Response

The reply to the referee is given in the attached pdf.

Reviewer 2 Report

The authors present results of statistical analysis of trajectory distribution functions for different tennis balls and net heights using GPU environment. I find the results interesting but it is not clear the contribution of the work. Furthermore, I believe that this work is similar to the following paper:

Schneider, R.; Kalentev, O.; Ivanovska, T.; Kemnitz, S. Computer simulations of table tennis ball 532 trajectories for studies of the influence of ball size and net height. International Journal of Computer Science in Sport 2013, 12(2), 25-35

I would like to suggest the authors to report the differences of the above research works.

I have also the following suggestions:

1) The tables in the paper should be formatted.

2) The authors report related work and an comphrensive comparison in order to understand the contribution of this work.

3) The authors would provide a list of concrete conclusions and contributions in a conclusion section.

Author Response

The reply to the referee is given in the attahced pdf.

Reviewer 3 Report

This is an interesting piece of work discussing effects due to the modifications of table tennis rules through analyzing numerical trajectory statistics. Especially, it discussed the effects from different table tennis ball radius and the net height from a large number of Monte Carlo simulations. From the simulation data, the results with 38-mm and 40-mm balls show similar statistical performance, while the 44-mm ball gives larger difference. I recommend to accept this paper with the following points addressed.

Most importantly, I feel more detailed should be added to offer a better illustration about the numerical algorithm and strategies used in this research. In Section 2, only a brief description of the  physics and several basic equations are shown. It would be helpful if the authors could provide more explanations about their method for solving the table tennis ball trajectory problem, and further the model assumptions and potential limitations in the numerical method used here. In this way, it can help the readers to get a better understanding about the strategies, and provide a more complete picture for the general idea.

Also for the analysis of the data and results, it would also be helpful if more detailed discussions can be offered, instead just a simple description of the listed data and figures. It appears from the PCA analysis, the 38-mm and 40-mm balls show similar statistics, while the 44-mm ball performs quite differently. Is there better physical interpretation of the leading PCA modes, and how can the differences in the identical or distinct results with different ball radius be interpreted?

Author Response

The reply to the comments is given in the attached pdf.

Round  2

Reviewer 1 Report

The title of this paper is "Statistical Analysis of Table Tennis Ball Trajectories," which mainly focusing on analysis. Therefore, the computational cost of GPU should not be a concern. Instead, accuracy should be the key consideration. In practice, many algorithms with irregular workload distributions have been successfully implemented on GPUs.  It is strongly recommended to conduct a subset of the current experiments for comparing Euler and RK4 methods.

The results reported in this paper and that in their previous work should be compared clearly to prove that the conclusions of this paper are really new contributions.

Author Response

Please find the reply in the attached pdf file

Reviewer 2 Report

The authors have addressed my previous comments appropriately.

Author Response

Thank you for supporting the publication by your refereeing.